Absence of heterosis in hybrid crested newts

http://orcid.org/0000-0003-3229-5993 Arntzen Jan W. 1 pim.arntzen@naturalis.nl
Üzüm Nazan 2
http://orcid.org/0000-0001-9115-6622 Ajduković Maja D. 3
http://orcid.org/0000-0002-6247-8849 Ivanović Ana 1 4
Wielstra Ben 1 5 6
1 Naturalis Biodiversity Center , Leiden , The Netherlands
2 Department of Biology, Faculty of Arts and Sciences, Adnan Menderes University , Aydin , Turkey
3 Institute for Biological Research “Siniša Stanković”, University of Belgrade , Belgrade , Serbia
4 Institute of Zoology, Faculty of Biology, University of Belgrade , Belgrade , Serbia
5 Department of Animal and Plant Sciences, University of Sheffield , Sheffield , UK
6 Department of Ecology and Evolutionary Biology, University of California , Los Angeles, CA , USA
Measey John
Electronic publication date: 2018 Jul 24
Publication date: 2018
Volume: 6
Electronic Location ID: e5317
Received 2018 May 15; Accepted 2018 Jul 4
Copyright: © 2018 Arntzen et al.
Copyright year: 2018
Copyright holder: Arntzen et al.
License: This is an open access article distributed under the terms of the Creative Commons Attribution License, which permits unrestricted use, distribution, reproduction and adaptation in any medium and for any purpose provided that it is properly attributed. For attribution, the original author(s), title, publication source (PeerJ) and either DOI or URL of the article must be cited.
License URL: https://creativecommons.org/licenses/by/4.0/

Keywords: Contact zone, Age distribution, Serbia, Hybrid zone, SNP markers, Skeletochronology

Funding: Marie Skłodowska-Curie grant agreement No 655487 Serbian Ministry of Education, Science and Technological Development grant no 173043 Naturalis Biodiversity Center B. Wielstra received funding from the European Union’s Horizon 2020 research and innovation programme under the Marie Skłodowska-Curie grant agreement No. 655487. M. D. Ajduković and A. Ivanović were supported by Serbian Ministry of Education, Science and Technological Development grant no. 173043. A. Ivanović was in receipt of a ‘Temminck grant’ provided by Naturalis Biodiversity Center. The funders had no role in study design, data collection and analysis, decision to publish, or preparation of the manuscript.

==============================
Relationships between phylogenetic relatedness, hybrid zone spatial structure, the amount of interspecific gene flow and population demography were investigated, with the newt genus Triturus as a model system. In earlier work, a bimodal hybrid zone of two distantly related species combined low interspecific gene flow with hybrid sterility and heterosis was documented. Apart from that, a suite of unimodal hybrid zones in closely related Triturus showed more or less extensive introgressive hybridization with no evidence for heterosis. We here report on population demography and interspecific gene flow in two Triturus species (T. macedonicus and T. ivanbureschi in Serbia). These are two that are moderately related, engage in a heterogeneous uni-/bimodal hybrid zone and hence represent an intermediate situation. This study used 13 diagnostic nuclear genetic markers in a population at the species contact zone. This showed that all individuals were hybrids, with no parentals detected. Age, size and longevity and the estimated growth curves are not exceeding that of the parental species, so that we conclude the absence of heterosis in T. macedonicus–T. ivanbureschi. Observations across the genus support the hypothesis that fertile hybrids allocate resources to reproduction and infertile hybrids allocate resources to growth. Several Triturus species hybrid zones not yet studied allow the testing of this hypothesis.

Introduction

Hybrid zones are regions where genetically distinct populations meet and hybridize (Barton & Hewitt, 1985; Harrison, 1993). They provide natural settings for the study of speciation. In particular they allow research on the consequences of new, previously untested genetic combinations of differentiated genomes in nature. Hybrid zones may take several forms, from long and narrow zones to large areas of overlap and mosaics (Arnold, 1992). The positive relationship between the degree to which hybridization proceeds and the phylogenetic relatedness of the species involved is well established (Jiggins & Mallet, 2000). In spite of some variation in the strength of reproductive isolation for individuals from spatially isolated populations, or among different species pairs from the same evolutionary lineages (Seehausen et al., 2014; Kearns et al., 2018) accumulated evidence suggests that, in diploid animals, reproductive isolation increases and introgression between lineages decreases with divergence (Singhal & Moritz, 2013; Arntzen, Wielstra & Wallis, 2014; Beysard & Heckel, 2014; Dufresnes et al., 2014; Montanari et al., 2014; Taylor et al., 2014). Unfortunately, much less is known about how species’ ecological preferences—and with that species distributions and local spatial configurations—may affect hybridization and vice versa, and how phylogenetic relatedness affects the interaction with ecology. Yet, such knowledge is relevant for the understanding of hybrid zones and the evolutionary inferences we draw from them. As Barton & Hewitt (1985: 121) proposed ‘… as soon as the loss of fitness through hybridization inside the zone becomes small enough … the zone will collapse into broad sympatry’. Accordingly, the more closely related hybridizing species are, the more hybridization will take place, over yet smaller areas. A negative relationship between the degree of hybridization and the amount range overlap was indeed noted by Zuiderwijk (1980) in a variety of European amphibian species pairs, but geographical variation and ecological differences remain to be studied.

We propose that a good group to investigate if reproductive isolation accumulates gradually would fulfil the following requirements. It would (i) be monophyletic and (ii) show more or less contiguous species ranges where (iii) closely as well as distantly related species engage in hybrid zones and (iv) ecological profiles of the species are different. Finally, (v) to reduce the impact of spatial scale a low dispersal capability would be an asset. One group that qualifies is the European newt genus Triturus. The genus is diverse and has a pan-European distribution, with outer ranges up to the Atlantic and the Mediterranean and reaching into Scandinavia, Russia, the Caucasus and Iran. The species group inner ranges form a patchwork and all nine species are, in one place or the other, involved in intra-generic hybridization (Arntzen, Wielstra & Wallis, 2014; Wielstra et al., 2014b). The genus Triturus is composed of three clades, namely the Triturus marmoratus group (or marbled newts) with two species (clade A), the T. cristatus group with four species (clade B) and the T. karelinii group with three species (clade C). Clades A and B engage in western Europe and clades B and C meet up in southeastern Europe (Arntzen, 2003; Wielstra et al., 2014b). See Fig. 1 for species distributions and phylogenetic relationships.

Figure 1 Distribution of nine Triturus species over Europe and the Near East.

Distribution of nine Triturus species over Europe and the Near East (after Wielstra & Arntzen, 2011). Major clades are (A) the marbled newts, (B) the T. cristatus species group of crested newts and (C) the T. karelinii species group of crested newts. Note that the spatial contact of clades is limited to central France (clades A and B) and the southern Balkans (B and C). Heterosis in size and longevity was observed in A × B hybrids (Mayenne, France, white dot, Francillon-Vieillot, Arntzen & Géraudie, 1990) and not in B × C hybrids (Vlasi, Serbia, asterisk, present paper). The insert shows the species phylogeny.

Clades A and B are distantly related with an estimated 27.6 Ma period of lineage independence (Wielstra & Arntzen, 2011). At their hybrid zone in France, T. marmoratus (clade A) and T. cristatus (clade B) interspecies F1 hybrids are infrequent (ca. 4% of the total population) and introgression is rare (<0.1%, Arntzen et al., 2009). Heterosis is the increased vigour in a cross between two genetically differentiated lines as compared with either of the parental lines. The T. cristatus–T. marmoratus F1 hybrids show hybrid vigour in body size and longevity, that is, they get older and larger than their parents (Francillon-Vieillot, Arntzen & Géraudie, 1990; Fig. 2A). The T. cristatus–T. marmoratus hybrid zone is bimodal, i.e. with predominantly the parental species and few hybrids, with a well-understood ecological differentiation. Forested and hilly areas are mostly occupied by T. marmoratus and open and flat areas have T. cristatus (Schoorl & Zuiderwijk, 1981; Visser et al., 2017). Representatives of clades B and C—together known as crested newts—show an intermediate level of phylogenetic relatedness with 10.4 Ma of lineage independence. European species involved in the contact are T. cristatus, T. dobrogicus and T. macedonicus in clade B and T. ivanbureschi in clade C. Among these four, T. dobrogicus stands out as a lowland species (Vörös et al., 2016). The species meet up in the Balkan peninsula along the lower Danube and in a wide zone running from Belgrade (Serbia) to Thessaloniki (Greece) (Fig. 1). Species within the B and C clades have 8.8–5.3 Ma of lineage independence and engage in a variety of more or less unimodal hybrid zones, i.e. with a majority of hybrids and few or no parentals (Arntzen, Wielstra & Wallis, 2014; Wielstra et al., 2017a, 2017b), with no evidence for heterosis so far. For data reviews in Triturus see Arntzen (2000) and Lukanov & Tzankov (2016). Because of the intermediate level of relatedness, the genetic and spatial interactions among species in the B and C clades are of special interest. We here studied hybridization between T. macedonicus and T. ivanbureschi where they meet in southeastern Serbia. A phenotypically mixed population was examined with a panel of nuclear genetic markers to estimate ancestry and heterozygosity and individual age was estimated by skeletochronology. We searched for heterosis by analysing hybrids for longevity, body size and growth.

Figure 2 Von Bertalanffy growth curves for Triturus newts.

(A) Triturus cristatus (orange), T. marmoratus (green) and Triturus cristatus—T. marmoratus F1 hybrids (black) from Mayenne, France. Data are from Francillon-Vieillot, Arntzen & Géraudie, (1990). Females are larger than males of the same age. The hybrids become larger and older than the parental species. No model could be fitted for T. marmoratus females. (B) Triturus macedonicus × T. ivanbureschi hybrid population from Vlasi in southeastern Serbia (black lines), T. macedonicus from Montenegro (pink) and T. ivanbureschi populations (blue) at localities Keşan and Klaros. Reference data are from Cvetković et al. (1996), Üzüm (2006), Üzüm & Olgun (2009) and Supplemental Information S6). Females are larger than males of the same age in three out of four populations. The Vlasi populations of hybrids shows no evidence for hybrid vigour in size, growth or longevity. The parameters that describe the Von Bertalanffy growth curves and the confidence intervals are presented in Supplemental Information S5.

Materials and Methods

Post-metamorphic crested newts were caught with funnel traps from March to July 2013 in a pond near the village Vlasi in the southeast of Serbia (43.00 N, 22.64 E, altitude 468 m a.s.l.). The focal newt population we henceforth refer to as ‘Vlasi.’ In total, 336 individuals were measured and marked and a small part of the newt’s tail at its tip was taken for genetic analyses (Arntzen, Smithson & Oldham, 1999). The population sample was classified in five groups determined from external morphology and colouration characteristics with documented phenotypes as a starting point (Wallis & Arntzen, 1989; Arntzen & Wallis, 1999; Arntzen, 2003; Wielstra et al., 2013b). Classes were as follows: group 1—T. macedonicus-like (N = 26), group 2—leaning towards T. macedonicus (N = 48), group 3—intermediate phenotype (N = 146), group 4—a phenotype leaning towards T. ivanbureschi (group 4, N = 83) and group 5—T. ivanbureschi-like (N = 33). Permission for fieldwork, for marking and to collect tissue samples was obtained from the Ministry of Energy, Development and Environmental Protection of the Republic of Serbia (permit no. 353-01-35/2013-08).

Molecular data were gathered for a panel of nuclear encoded SNP markers (see below). We included reference samples with N = 3 for seven populations of T. ivanbureschi and seven populations of T. macedonicus available from Wielstra et al. (2013a) (Table 1). Reference populations were located to the south of the Vlasi population, distant from other, more northerly distributed Triturus species. Wielstra et al. (2014a) produced sequence data for 52 short (ca. 140 bp) nuclear markers positioned in 3′UTR regions of protein-coding genes for three individuals from four populations positioned throughout the ranges of both T. ivanbureschi and T. macedonicus. We focussed on the subset of 24 nuclear markers with species diagnostic allele variants for T. ivanbureschi and T. macedonicus. Additionally an mtDNA SNP (nd4) was designed from Sanger sequence data taken from Wielstra et al. (2013a). We determined diagnostic SNPs by checking the sequence alignments by eye in MacClade 4.08 (Maddison & Maddison, 2005).

Table 1 Populations of Triturus macedonicus, T. ivanbureschi and the mixed focal population from Vlasi, Serbia, genotyped with a panel of nuclear genetic markers.

Taxon, locality, country	Coordinates	Population used for marker design	Sample size	Tissue collection number	
Latitude	Longitude	
Triturus macedonicus						
 Paliambela, Greece	38.909	20.970		3	2815-7	
 Kerameia, Greece	39.562	22.081	Yes	3	3775-7	
 Kounoupena, Greece	39.683	19.764	Yes	3	2820-2	
 Lushnjë, Albania	41.000	19.664		3	3472-4	
 Vrbjani, Macedonia	41.413	20.816		3	3327, 3583-4	
 Gostivar, Macedonia	41.817	20.899	Yes	3	3601-3	
 Bjeloši, Montenegro	42.374	18.907	Yes	3	3245-7	
Triturus ivanbureschi						
 Shumnatitsa, Bulgaria	42.297	23.626		3	2779-81	
 Saint Kosmas, Greece	41.084	24.669		3	2846-8	
 Alexandrovo, Bulgaria	42.601	25.093	Yes	3	2492-4	
 Keşan, Turkey	40.917	26.633	Yes	3	2360-2	
 Salihler, Turkey	39.181	26.826		3	1808-9, 1812	
 Alepu, Bulgaria	42.348	27.714	Yes	3	2602-4	
 Kocabey, Turkey	39.352	28.217	Yes	3	1879-81	
Focal population						
 Vlasi, Serbia	42.998	22.637		336	6284-6620	
Note:

Tissus are kept at Naturalis Biodiversity Center (https://www.naturalis.nl/en/).

Genotyping was conducted commercially at the SNP genotyping facility of the Institute of Biology, Leiden University, using the Kompetitive Allele-Specific PCR (KASP) genotyping system (LGC KBioscience, Teddington, UK). This involves fluorescence-based genotyping using SNP-specific primers. We used the program Kraken to design two forward or reverse primers that are specific for (i.e. with a final base complementary to) one of the two potential SNP variants, in addition to a general reverse or forward primer (Semagn et al., 2014). We genotyped 378 crested newts in total: 21 individuals of both parental species (the eight populations on which SNP identification was based plus an additional six populations) and 336 individuals from Vlasi. Sequence alignments and Kraken input for all markers are available in Supplemental Information S1. Raw output of the KASP genotyping protocol is in Supplemental Information S2. The Ion Torrent next-generation sequence data of Wielstra et al. (2014a) used for SNP discovery are available from Dryad Digital Repository entry DOI 10.5061/dryad.36775. The genetic and morphological data used in the analysis are presented in Supplemental Informations S3 and S4.

The program NewHybrids (Anderson & Thompson, 2002) was used to estimate the proportion of the Vlasi population consisting of recently formed hybrids. Reference populations were invoked with the ‘z-option’, otherwise settings were default. The program HIest was used to estimate heterozygosity (H) and ancestry (S) (Fitzpatrick, 2012). To evaluate the consistency and independence of the genetic data and to assess hybrid zone structure Hardy–Weinberg equilibrium (HW) and linkage disequilibrium (LD) were quantified with GenePop (Raymond & Rousset, 1995).

All animals encountered were marked by clipping the middle toe on the right foot for the purpose of a population size estimate by the capture-recapture method, with results not here reported. In a subsample of individuals (N = 170, 77 males, 85 females and eight small individuals with secondary sexual characters not expressed, i.e. juveniles) that represented the five morphological groups, a phalanx of the toe was taken for age determination by skeletochronology. Lines of arrested growth (LAGs) were counted as in Francillon-Vieillot, Arntzen & Géraudie (1990). Endosteal resorption was estimated by comparing the diameters of eroded marrow cavities of adults with the diameters of non-eroded marrow cavities of juveniles and was observed in 75 males (97.4%), 72 females (91.7%) and two juveniles. Individual ages were estimated taking endosteal resorption into account. We also determined age at maturity from ‘rapprochement’. This is the tightening of LAGs that is associated with the shift of resources from growth to reproduction. In some cases we observed a line formed at metamorphosis. Some examples are in Fig. 3.

Figure 3 Cross-sections of a phalanges of Triturus macedonicus × T. ivanbureschi crested newt from Vlasi.

Cross-sections of a phalange of a juvenile, male and female Triturus macedonicus × T. ivanbureschi crested newt from Vlasi. Study sites: m.c., marrow cavity; m.l., metamorphosis line; e.r., endosteal resorption; p., periphery; rap., rapprochement. Note that the periphery is not regarded as a LAG. Sections are 18 μm thick and taken at the level of the diaphysis. (A) Three-year-old juvenile, SVl 48 mm. Three LAGs (arrow heads) are observed in the periosteal bone as well as a metamorphosis line (arrow). (B) Five-year-old male, SVl 64 mm. Four LAGs are observed in the periosteal bone (arrow heads). Endosteal resorption present (arrow). (C) Six-year-old female, SVl 64 mm. Five LAGs are observed in the periosteal bone (arrow heads). Endosteal resorption and rapprochement are also marked by arrows.

Body size was measured in millimetre from the tip of the snout up to the posterior end of the cloaca (snout vent length, SVl). Sexual dimorphism was estimated with the Lovich & Gibbons (1992) sexual dimorphism index SDi = (mean length of the larger sex/mean length of the smaller sex) minus unity, and arbitrarily defined as positive when females are larger than males and negative in the reverse case. Von Bertalanffy growth curves were determined with the function ‘growthmodels’ in the R package FSA (R Core Team, 2013; FSA, 2017). Since the morphological groups that we distinguished showed no marked genetic differentiation (see below) the data on size, age and growth were pooled for both sexes. The few juveniles in the sample remained unsexed and were excluded from the analyses. Confidence intervals were estimated by bootstrapping with nlstools in 2000 bootstrap iterations (Nlstools, 2015). Differences between morphological groups in the level of estimated ancestry and heterozygosity were tested by ANOVA. The relationship between level of heterozygosity and SVl was tested by the Pearson correlation coefficient. ANOVA and correlations were done with SAS software (SAS Institute Inc., 2011).

Results

Out of the candidate nuclear markers two (cnppd and kdm3) were dropped because a diagnostic SNP could not be identified and for four (ace, eif4ebp2, ssh2 and syncrip) the Kraken software could not design a suitable set of primers. For the 18 remaining diagnostic nuclear markers for which assays could be designed, PCR amplification for one (amot) failed. The locus slc25 yielded a high frequency of missing data (16%) in the reference populations, suggesting it did not amplify the T. ivanbureschi-allele well and results for this marker were also discarded. For the remainder missing data amounted to 1.8%. The four loci ahe, ddx17, dnaj and sre showed highly significant LD for all six pairwise combinations. This result was interpreted as tight physical linkage in a single linkage group. To increase data independence results for the locus ahe (with the fewest missing data) were retained and the others dismissed. Subsequent tests for HW and LD yielded no significant comparisons under Bonferroni correction. The KASP genotyping output for the remaining 13 markers is summarized in Supplemental Information S3. One nuclear SNP marker (col18) showed a single instance of heterozygosity in a parental individual, suggesting either a genotyping error or a marker that is not fully diagnostic. With N = 82 (21.7%) of missing data the mtDNA marker (nd4) performed relatively poorly. All newts from the Vlasi population that could be genotyped possessed the mtDNA haplotype typical for T. ivanbureschi, as expected for this marker in this system (for background information see Wielstra et al., 2017a).

In NewHybrids none of the individuals from the Vlasi population were allocated to either of the parental species, to the F1 hybrid class or to the class of backcrosses towards T. macedonicus. Several individuals were allocated to the ‘backcross to T. ivanbureshi class’ (N = 8, 2.4%). The majority of individuals was classified as F2 hybrids (N = 328, 97.6%). When NewHybrids was instructed to take not two but three generations into account, the majority (99.7%) was allocated to that third generation. HIest grouped all Vlasi individuals in the middle of the ancestry times heterozygosity bivariate plot, somewhat off-centre in the direction of T. ivanbureschi (Fig. 4), with the reference populations in the lower left (pure T. macedonicus) and lower right corners (pure T. ivanbureschi). To explore possible differences between morphological groups in the level of estimated ancestry (S) and heterozygosity (H) ANOVA was used over five (groups 1–5) and three morphological groups (groups 1, 2–4 pooled, 5) with no statistically significant results. Pearson correlation coefficients for S and SVl and H and SVl were also not significant, for males as well as for females.

Figure 4 Ancestry versus heterozygosity plot based on 13 species diagnostic nuclear genetic markers.

Individuals from the Vlasi population are shown by open round symbols (N = 336). The left corner of the triangle corresponds to Triturus macedonicus (solid square symbols, N = 21), the right corner to T. ivanbureschi (solid triangle symbol, N = 21) and the upper corner to (non-observed) F1 hybrids populations. The Vlasi populations of hybrids shows no evidence for hybrid vigour in size, growth or longevity.

The body size distribution showed that females were significantly larger than males (SVl males 63.9, SVl females 67.5 mm; Student’s t-test P < 0.0001; SDi = 0.056). The age distribution for males and females showed no significant difference (Mann–Whitney U-test, P > 0.05; Table 2). In both sexes >70% of the individuals had an estimated age of 5, 6 or 7 years. Age at maturity estimated by rapprochement was mostly 2, 3 or 4 years in both sexes (average males 3.1, N = 75; average females 3.0, N = 74; Supplemental Information S4.). The absence of an observable rapprochement was more frequent in females (12.9%) than in males (2.6%) (P < 0.05, G-test of independence), perhaps suggesting a less drastic shift in resource allocation in the transition from growth to reproduction in females. Longevity was 13 years in males and 11 years in females. The parameters that describe the Von Bertalanffy growth curves and their confidence intervals are presented in Supplemental Information S5. Results for the Vlasi populations were compared with data from the literature (Cvetković et al., 1996 and Supplemental Information S6 on T. macedonicus and Üzüm, 2006 and Üzüm & Olgun, 2009 on T. ivanbureschi). In three out of four populations females are larger than males of the same age. The other growth curves for populations and species are similar, with no more than 10 mm difference in SVl among the most different groups (Fig. 2B). Unlike T. cristatus × T. marmoratus hybrids, the T. macedonicus × T. ivanbureschi hybrids showed no heterosis. Longevity was higher in T. macedonicus than in T. ivanbureschi, with interspecific hybrids from the Vlasi population taking an intermediate position.

Table 2 Age distribution of juveniles, males and females in the admixed Triturus macedonicus–T. ivanbureschi population from Vlasi, Serbia.

Age	Juveniles	Males	Females	
0				
1				
2	2			
3	5		3	
4	1	6	5	
5		18	16	
6		20	24	
7		18	20	
8		9	11	
9		4	3	
10		1		
11			3	
12				
13		1		
				
Total	8	77	85	
Note:

Ages are estimated by skeletochronology.

Discussion

We encountered a population of crested newts near Vlasi in southeastern Serbia that showed extensive phenotypic variation and—being close to the territory of both T. macedonicus and T. ivanbureschi—a hybrid nature was assumed. We were, however, unable to determine the extent of hybridization from morphological and colouration characters and yet wanted to find out if this hybrid zone population has a unimodal character (with a majority of hybrids and few or no parentals), a bimodal character (with predominantly the parental species and few hybrids), or something in between. We were also interested in fitness consequences that hybridization might have on the demography of the population, in particular if heterosis would be combined with hybrid sterility.

All individuals from the focal population were allocated to the second generation of hybrids, indicating strong admixture along with the absence of recent crossings between the parental species. When a third generation was an option, individuals where allocated to that third generation, further supporting the absence of recent hybrids. Looking yet deeper into the coalescence would require more genetic markers. An ancestry versus heterozygosity plot (Fig. 4) further support that genotypes in this Vlasi population result from many generations of admixture. In the absence of selection and excluding physical linkage LD halves every next generation and will barely be measurable after a few generations (Barton & Gale, 1993). Accordingly, the absence of admixture LD also suggests that hybridization between T. macedonicus and T. ivanbureschi has been ongoing for more than a couple of generations. We conclude that Vlasi is a population with only hybrids and the parental species absent. A hybridized population was also suggested by the absence of genetically diagnosable morphological groups. These results suggest that T. macedonicus and T. ivanbureschi engage in a unimodal hybrid zone. The genetic affiliation of the Vlasi population is somewhat closer to T. ivanbureschi than to T. macedonicus (Fig. 4). This is in line with the documented distributions of the two species where the Vlasi population is geographically closer to T. ivanbureschi than to T. macedonicus (Wielstra et al., 2014b). However, the exact position of the centre and the width of the cline remain to be documented. This result contrasts with the situation to the northwest, where the two species engage over a wide area of species replacement. The mosaic distribution includes a T. ivanbureschi enclave that was cut off from its main distribution by superseding T. macedonicus (Arntzen, 2003; Arntzen & Wallis, 1999, cf. Fig. 1). The sole transmission of the mitochondrial haplotype typical for T. ivanbureschi, that we here confirmed for the Vlasi hybrid population, helped to reconstruct a scenario in which T. macedonicus advanced at the expense of T. ivanbureschi (Wielstra et al., 2017a).

Narrow zones with extensive hybridization are found at contacts between crested newts species belonging to clades B and C (Arntzen, Wielstra & Wallis, 2014; Wielstra et al., 2017b, cf. Fig. 1). A relationship appears to exist with species relatedness, so that closely related species form clines and more distant species engage in a mosaic distribution, as described for T. cristatus and T. marmoratus. The presence of a unimodal (clinal) as well as bimodal (mosaic) hybrid zone structure in the T. macedonicus–T. ivanbureschi contact is in line with an intermediate level of relatedness.

Conclusion

As demonstrated here, the T. macedonicus–T. ivanbureschi hybrids are fertile, producing new genetic combinations that themselves reproduce in a more or less narrow hybrid zone. This result for a comparison of the B and C clades contrasts with the results obtained earlier for T. cristatus and T. marmoratus in the A and B clades, where hybridization in is an evolutionary dead-end. These hybrids are largely sterile and introgression is near-absent (Arntzen & Wallis, 1991; Arntzen et al., 2009). We propose that in the genus Triturus heterosis is restricted to hybrids that are infertile and that—also in adult stage—direct their resources more to growth than to reproduction. This is more likely to be the case for genetically incompatible than for genetically compatible species. The testing grounds for this assertion are several unimodal hybrid zones of closely related Triturus species, i.e. within the A and B clades (Fig. 1) where demonstration of heterosis would contradict our hypothesis.

Supplemental Information

Supplemental Information 1 Input consensus sequences for Kraken.

Click here for additional data file.

Supplemental Information 2 Raw output of the KASP genotyping protocol.

Click here for additional data file.

Supplemental Information 3 Genetic data for a hybrid newt population in Serbia.

Click here for additional data file.

Supplemental Information 4 Skeletochronology results along with morphometric data and individual Structure scores.

Click here for additional data file.

Supplemental Information 5 Parameter values and 95% confidence intervals (CI) in the Von Bertalanffy growth curves for Triturus populations.

Parameter values and 95% confidence intervals (CI) in the Von Bertalanffy growth curves for Triturus populations in the west and southeast of the genus’ range. Triturus cristatus and T. marmoratus are hybridizing in a wide area of range overlap. The focal Vlasi population is of mixed T. macedonicus–T. ivanbureschi ancestry.

Click here for additional data file.

Supplemental Information 6 Skeletochronology results for Triturus macedonicus.

Skeletochronology results for Triturus macedonicus from three populations in Montenegro, located away from areas in interspecific hybridization. The material is from the batrachalogical collection of the Institute of Biological Research “Siniša Stanković” and was collected in 1993–1996 (Džukić et al., 2015). Afterwards the material was cleared and stained for other studies (Ivanović et al., 2008, Tomašević Tomašević-Kolarov, Ivanović & Kalezić, 2011, Ivanović & Arntzen, 2014).

Click here for additional data file.

We thank R. Bancila and F. Stanescu for help with curve fitting statistics, O. Schaap and K. Vrieling for running the SNP-line and three anonymous reviewers for helpful comments.

Additional Information and Declarations

Competing Interests

Author Contributions

Animal Ethics

Data Availability

The authors declare that they have no competing interests.

Jan W. Arntzen conceived and designed the experiments, analysed the data, contributed reagents/materials/analysis tools, prepared figures and/or tables, authored or reviewed drafts of the paper, approved the final draft.

Nazan Üzüm performed the experiments, analysed the data, approved the final draft.

Maja D. Ajduković performed the experiments, approved the final draft.

Ana Ivanović conceived and designed the experiments, analysed the data, contributed reagents/materials/analysis tools, authored or reviewed drafts of the paper, approved the final draft.

Ben Wielstra conceived and designed the experiments, performed the experiments, analysed the data, contributed reagents/materials/analysis tools, authored or reviewed drafts of the paper, approved the final draft.

The following information was supplied relating to ethical approvals (i.e. approving body and any reference numbers):

The Ministry of Energy, Development and Environmental Protection of the Republic of Serbia (today, Ministry of Environmental Protection, Agency for Nature Protection) provided full approval to the Institute of Biological Research “Siniša Stanković” to conduct research on the hybrid population (Vlasi) and for DNA tissue sampling. The permission is administrated as No. 353-01-35/2013-08.

The following information was supplied regarding data availability:

The raw data are provided in the Supplemental Files.

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
