# Peer review of "Absence of heterosis in hybrid crested newts"

_PeerJ, doi:10.7717/peerj.5317_

## Round 0.1 · original submission · Minor Revisions

Your ms has now been reviewed by 3 referees, all of whom suggest that only minor revisions are needed before this ms can be accepted. Having read you ms, I agree with this opinion and now direct the authors to make these revisions.

I agree with reviewer 1 that you should give relevant details of numbers of individuals used in the study. There are some ambiguities in the ageing methodology that could be easily clarified. However, I don't think that the skeletochonology aspect of the study needs to be expanded out of context.

I also agree with reviewer 2 that there are numerous places where the ms is not clear to a reader not familiar with the models and the existing literature. It should be possible to read and make sense of this paper in isolation of the other reports.

Reviewer 3 points out that the abstract leads the reader to think that the conclusions of this study are far more wide ranging than they are. Please re-write with the results of this study only in mind, and make it clear when you are referring to previously published studies.

Overall, as you'll note in my comments on the pdf, I found that the manuscript was at times hard to follow. This is particularly with respect to distinguishing this study from previous studies. I don't think that this will require any major effort on your part. Thanks for your submission.

Reviewer 1 ·

Basic reporting

Language
The language used is American English, which is clear and unambiguous throughout the manuscript. On line 132 the following misspelled word was noted: “growh” instead of “growth”.

Intro & background
The research background is well presented and the cited literature is relevant. However, there are issues with one of the references:
A reference from the list is missing from the main text of the manuscript: Arntzen, J. W., & Wallis, G. P. (1991). Restricted gene flow in a moving hybrid zone of the newts Triturus cristatus and T. marmoratus in western France. Evolution, 45, 805-826 (lines 258, 259).

Structure
The structure of the manuscript conforms to the PeerJ standards.

Figures
The figures are relevant, in high quality, well labelled and described. However, the manuscript will benefit from an additional figure, showing a stained cross-section of a phalange with arrows indicating LAGs, age at maturity and endosteal resorption (and possibly line of metamorphosis).

Raw data
The raw data used for the analyses is supplied.

Experimental design

Originality
The manuscript presents original primary research that is a continuation of and addition to a series of studies on Triturus phylogeny, conducted by the authors. The research falls within the scope of PeerJ.

Research question
The research question is well defined and relevant, aimed at filling a knowledge gap for the hybridisation process in the genus Triturus.

Investigation and methods
The investigation meets accepted ethical standards and for the most part is performed to a high technical level. Most methods are described with sufficient detail and information in order to be replicated. Where the Materials and methods section is lacking is in the description of the skeletochronology (lines 130-139), which is very superfluous and without any detail. Firstly, they fail to state the exact number of individuals from each morphological group – from what can be counted in the raw data (peerj-28183-SupplInf_4.xls) these are approximately half of each group, and the reason for this decision must be explained. Secondly, the statement “Lines of arrested growth (LAGs) were counted as in Francillon-Viellot et al. (1990)” (line 132) is ambiguous – if the authors followed the exact procedure of Francillon-Viellot et al. (incl. cross-section thickness, preparation and staining methodology), they should state so more clearly; if not, provide information on how the cross-sections were actually prepared. The authors should also clearly state how they have interpreted and controlled for endosteal resoprtion – this process could potentially destroy the first LAGs and if overlooked could skew the results of age determination (i.e. estimated age could be 2-3 years less than the actual age). In addition, the “Other statistical analyses” (line 139) should be clearly stated and their use justified.

Validity of the findings

For the most part the data presented by the authors is robust and the statistical analyses have been adequately interpreted. Still, some issues are left unexplained:
1. What was the observed level of endosteal resorption and how many LAGs were affected? This is important, as the authors themselves state that increased longevity is one aspect of the hybrid vigour and ignoring resorption could lead to erroneously low estimated age.
2. Was a metamorphosis line observed? Its presence could be related to harsher conditions and, consequently, to longevity and growth. Since some of the authors have publications on estimating age with the presence of metamorphosis line and endosteal resorption (e.g. Üzüm & Olgun 2009), there is no need to recommend reference literature – this is something they know how to do and need to do it in order to better validate their findings.
3. Were there statistically significant differences in terms of SVL, age or longevity between the five morphological groups from Vlasi? Even if genetically they all classify as second generation hybrids, this data could still be informative.
4. The authors state that females are larger than males – but for the purpose of better comparison to other members of the Triturus genus, a sexual dimorphism index could be calculated (following Lovich & Gibbons 1992, as some of the authors have done in previous publications – e.g. Üzüm & Olgun 2009).
5. The authors state that “Longevity increased over species as T. cristatus, T. marmoratus, T. ivanbureschi and T. macedonicus (Figures 2 and 4).” (lines 182, 183). However, from figure 4 it is clear that longevity of T. macedonicus is greater than that of the T. macedonicus x T. ivanbureschi hybrids. Furthermore, in the text of figure 4 it is stated that “The Vlasi populations of hybrids shows no evidence for hybrid vigour in size, growth or longevity.”

Additional comments

Overall, the manuscript is informative and will present a valuable contribution towards understanding hybridisation in closely related amphibian species, newts in particular. However, it cannot be published in its current form for the following reasons (ordered by importance):
1. It is almost certain that endosteal resorption was present in at least part of the studied individuals. This process should be discussed and carefully controlled for, because if the actual longevity of the hybrids is greater than the authors have estimated, this would dent their conclusion on the absence of heterosis in T. macedonicus x T. ivanbureschi hybrids.
2. Although less important, the line of metamorphosis should also be discussed, as if it is present in a large number of individuals, this could indicate harsher conditions that negatively affect all tested heterosis components (size, growth and longevity).
3. A presence of figure of a stained cross-section with the main features indicated by arrows is next to mandatory in a serious skeletochronology study.
4. Overall, the data from the skeletochronology and the morphological measurements should be better presented and discussed (see points 3-5 from Validity of the findings).

Reviewer 2 ·

Basic reporting

Some editorial work needed, odd and awkward phrasing is present.

Experimental design

Seems fine.

Validity of the findings

Interpreted reasonably.

Additional comments

Dear Editors at PeerJ,

This paper identifies hybrid individuals of two species using genetic markers and then assesses growth curves, finding that there is no difference between the hybrids and parental species. I think the paper is methodologically sound, and results soundly interpreted. I do think that there needs to be a more careful explanation of the lack of statistical differences for the main result that growth curves are not different between the hybrids and the parental species (more details below). I also think that the discussion need to highlight that the interpretation of a "unimodal hybrid zone" (line 204) needs to qualified in that only a single pond was assessed. Perhaps this is the core of the hybrid zone where hybrids are abundant, but elsewhere in the areas of sympatry they are not. Basically, it's not robust to say that the whole hybrid zone in unimodal, when only a single locality in that zone was assessed. Other than that, I think there are some clarification (awkward wording/phrasing) and methodological details that need to be made.


Specific comments:

Abstract sentence 2 I don't understand: "A bimodal hybrid zone of distantly related species in France combined low gene flow with hybrid sterility and heterosis whereas a suite of unimodal hybrid zones in closely related Triturus showed more or less extensive introgressive hybridization with no evidence for heterosis." Is this in reference to other studies?

Abstract L27 & 29: "The study with..."; should be "This study used 13". Also, the next sentence has awkward phrasing, and should probably be something like "estimated growth curves are not different from the parental species".

L43: "evidence accumulates that", should be "accumulated evidence suggests that"

L97: confusing language.

L96--99: How were multiple phenotypic measures used to make these designation decisions? Was a PCA performed? Is there some quantifiable metric that is used/scoring card scheme?

L103--104: Are those known to be pure populations? How?

L105: "to stay away from", bad phrasing. Should probably be something like "distant from other species"

L134: How are those 13 individuals distributed across the age and sex classes (i.e., all juveniles)? How was a juvenile classification made (if they are sexually mature, i.e., have rapprochment, then they are not juvenile)?

L137: please cite the R core team as well.

L125: I think you should say why you tested HWE and LD (presumably to look for the presence of null alleles and that markers were unlinked...no?)

L128: What sort of prior did you set (Jefferies, or uniform)? Did you test if the prior had an effect (as suggested in the NewHybrids manual)? Did you define allele frequencies in the fashion described in the manual (section 3.5.2 is relevant to your markers, I think, as it sounds like your markers are limited to one species or the other.)

L132--134: How was this rapprochment decided? Is there some sort of quantifiable measurement?

L145: What constitutes a "high frequency"? Please clarify.

L151: change to "significant comparisons"

L155: Ambiguous which marker these results are about, presumably the mtDNA? Please clarify. Also, please state here why this is expected, so the reader does not have to go dig up another paper Also, how was this determined that the offspring had this marker of the one species, by eye, with a tree, blast, etc.?

L165: Please clarify what H and S refer to, and how this ANOVA was done (presumably in the HLsest software?).

L167: Could you please add a sentence describing why this is important and what this indicates? (for both the anova test, and the correlation).

L169: Was this limited to hybrids? Or all samples?

L174: Please clarify what you mean by "a more regular transition". Do you mean a less drastic shift in resource allocation?

**L175--183**: It is not clear that there was a statistical test done here. It seem that in table 3, all of the CIs are overlapping (it would be nice to have that in the growth curve plot), and that is why you say there is no hybrid vigor. Perhaps a statement attesting to the statistical nature of this test needs to be done. I also think a more general test could be done with these data, like a multiple regression with age and size as dependent variables, and species (macedonicus, ivansbureschi, and hybrid) and sex as independent (the result of a non-significant species effect would support the conclusion). Or just a linear model with SVL as the response and age, species, and sex and the independent. But that is probably unnecessary, as table 3 seems to have equivalent information (people are just generally more familiar with linear models, I find). It could be helpful to have the CIs plotted along with the growth curves, if possible.

L188: Unclear wording. You say "we were unable to determine the extent of hybridization", but where was that information? The only morphological measurements I can see in this paper related to the growth curves (and is that what you mean? If so, perhaps reference that figure here).

L189--191: These definitions of unimodal or bimodal are clarifying, and should maybe be stated in the introduction (where I was uncertain what you meant by uni/bimodal hybrid zones).

L199: perhaps you mean "binning" instead of "banning"? I'm also uncertain what that refers to. Please clarify.

L200: citation needed.

L201: It's not clear what you mean by absence of LD. I thought your LD test was just between markers to ensure that you are not pseudoreplicating the genetic data (i.e., markers are independent). Please clarify.

L225: To be fair, you don't explicitly test here that hybrids are fertile, you only infer it from the presence of F2 or later hybrids. This sentence should probably be changed to reflect that.

L227: Citation needed, as you didn't test that here.

L231: Perhaps you should tag on that this would be an area for future investigation, because it's a little odd to end your paper saying what could contradict it without saying why.

Figure 2/4: I think it would be helpful to group figures 2 and 4 into a single figure (as "a" and "b" subfigures), to show what heterosis would look like, if it were to be present.

Figures 1--4: Y axis labels should be next to the Y axis, not on top.

Reviewer 3 ·

Basic reporting

This study reports on the hybrid origin of a single population of Triturus newts from Serbia. Based on diagnostic SNPs, the authors find that all individuals are at least second or third generation hybrids. They also estimate age, size and longevity in this population and reference populations of the parental species, and conclude that there is no heterosis in the hybrids (i.e. being fertile, the hybrids allocate energy into reproduction). The authors then contrast this hybrid population with data from T. cristatus/marmoratus from a previous study. The introduction provides some relevant information for understanding hybrid zones, and the aims of the paper are clearly defined.

Experimental design

The number (13) of diagnostic SNPs is rather small, but sufficient to determine the hybrid nature of the newt individuals. The analyses are well-done and clearly explained, the English is mostly unambiguous (see a few minor comments below), and illustrations & tables provide important information and are easy to understand.

Validity of the findings

I found one major issue which should be improved upon before acceptance – the validity of the main conclusion of the paper, mentioned in the abstract & conclusion sections. The authors propose that in the genus Triturus heterosis is restricted to hybrids that are infertile, and that this is more likely the case for genetically incompatible as opposed to genetically compatible species. This far-reaching conclusion is based on a single hybrid population analysed in this paper, and references to the cristatus-marmoratus hybrid zone and hybrid populations of some other species (which ones?) of the complex. No data is provided for these other hybrid zones (with the exception of some previously published data for Von Bertalanffy growth curves for marmoratus – cristatus). The analysis of other zones in the complex is thus a purely verbal one, and rather minimal at that. Therefore I find it hard to justify some of the statements in the abstract (e.g. first and last sentences), or the conclusions of the paper (lines 225-231). Indeed, the first sentences of the abstract (lines 20-24) had me thinking that the paper is an empirical analysis of several hybrid zones in the Triturus species complex. This is certainly not the case – there is data for a single hybrid population between two species.
I would recommend de-emphasizing this aspect of the paper and sticking to the main aims which are nicely laid out in lines 85-90. In particular, I would rework the abstract to more accurately reflect the data and analyses in the paper, and I would remove/modify the conclusions section which seems rather far-fetched given the data at hand. The idea that hybrid zone structure is at least partially determined by genetic divergence of the hybridizing species is interesting and worth pursuing in the discussion. However, I don’t think you can conclude that this is the case based on a single hybrid population and verbal treatment of a handful of other examples from the genus.

Additional comments

Minor issues:
Line 43 awkward wording, try: …, accumulated evidence shows that in diploid animals, reproductive isolation….

Lines 51-53. What exactly do you mean by level of hybridization? Spatial extent (width of hybrid zone), proportion of population in hybrid matings? Also, Zuiderwijk (1980) seems rather outdated, as many taxonomic changes have occurred in this vertebrate group since 1980.

Line 56. the accumulation of reproductive isolation is not really tested in this paper, therefore I suggest rewording the sentence by replacing reproductive isolation with introgressive hybridization, or something more general like speciation continuum, etc.

Line 95 Remove “A” (start sentence with Morphological)

Line 96, change to: determined five groups, including…

Line 100, collect tissues/tailtips

Lines 163-167, define the abbreviations H, S and SVl in this section

Line 203, replace “just” with “only”

Line 203, “the“ documented distributions

Line 215, should be “infrequent hybridization”

Line 225, should be “demonstrated here”

---

## Round 0.2 · accepted · Accept

Hello Pim

Thanks for your corrections, which I think help clarify and improve the text. On reading again, I note a few grammatical difficulties (highlighted in yellow in the attached). I'm not fond of dashes in sentences instead of brackets, but note that they are grammatically recognised. However, please make sure that they are the correct "em-dash" (—) and not a hyphen (-) as used.

For the figures:
Fig 1 - please make specific mention of the * and ? in the figure legend.
Fig 2 - I also find the placement of "Snout-vent length (mm)" ambiguous. It isn't clear whether it only refers to A, and there's no apparent association with the y-axis

Table 1 - can you mention in the legend where the tissue collection is held? I can't find this information anywhere in the document.

#